# Chemical Memory with Discrete Turing Patterns Appearing in the Glycolytic Reaction

**DOI:** 10.3390/biomimetics8020154

**Published:** 2023-04-13

**Authors:** Jerzy Gorecki, Frantisek Muzika

**Affiliations:** Institute of Physical Chemistry, Polish Academy of Sciences, Kasprzaka 44/52, 01-224 Warsaw, Poland

**Keywords:** chemical computing, memory, oscillations, discrete Turing pattern, glycolytic reaction

## Abstract

Memory is an essential element in information processing devices. We investigated a network formed by just three interacting nodes representing continuously stirred tank reactors (CSTRs) in which the glycolytic reaction proceeds as a potential realization of a chemical memory unit. Our study is based on the 2-variable computational model of the reaction. The model parameters were selected such that the system has a stable limit cycle and several distinct, discrete Turing patterns characterized by stationary concentrations at the nodes. In our interpretation, oscillations represent a blank memory unit, and Turing patterns code information. The considered memory can preserve information on one of six different symbols. The time evolution of the nodes was individually controlled by the inflow of ATP. We demonstrate that information can be written with a simple and short perturbation of the inflow. The perturbation applies to only one or two nodes, and it is symbol specific. The memory can be erased with identical inflow perturbation applied to all nodes. The presented idea of pattern-coded memory applies to other reaction networks that allow for discrete Turing patterns. Moreover, it hints at the experimental realization of memory in a simple system with the glycolytic reaction.

## 1. Introduction

Memory is an essential element in information processing devices. In the von Neumann computing architecture [1,2], it plays as important a role as the computing unit because information processing is based on a sequenced dataflow between memory and processor. The unprecedented success of semiconductor technology reflected by the Moore-type law for processors [3] is mirrored by a similar increase in the density of semiconductor memory units inside a single chip as the function of time [4].

One can hardly claim that the progress of chemical computing is as fast as the semiconductor one. Observation of Nature shows the enormous potential of information processing based on chemical reactions. Animals and humans are able to process a huge amount of information and solve complex problems at low energy cost with their nervous systems and brains that obviously work using chemical reactions [5]. Contrary to the achievements of Nature, man-made chemical information processing is still at the level of infancy [6,7]. The investigated systems are simple and perform elementary information processing operations [8,9,10,11].

Various chemical phenomena leading to multiple stable states can be used to construct memory devices. The most popular are self-organizing chemical reactions or reaction networks, where the reactions and transport between reagents give rise to complex activity patterns. Even in simple reaction–diffusion systems, the set of stable patterns can be rich. It was demonstrated that all capital letters of the English alphabet could be mapped to patterns in two-dimensional systems with zero-flux boundary conditions [12]. Structures corresponding to different characters were obtained within the same reaction model, but the initial conditions and dimension of the system were different for each letter. Another study illustrated a reaction–diffusion system where stable patterns imitated Hebrew letters [13].

Spatiotemporal patterns appearing in chemical systems can also be used for symbol coding. The observation that a pulse of excitation can rotate in a ring-shaped channel for a long time if the reactants are continuously supplied and the products removed [14] strongly influences the construction of chemical memory with a reaction–diffusion medium. A ring of an excitable medium can be regarded as a memory cell capable of storing one bit of information represented by a rotating pulse. If there is a rotating pulse, the state of memory corresponds to the logical “true” (the memory is “loaded”). If there is no such pulse, the memory state corresponds to the logical “false”. The idea of a ring as a chemical memory was investigated by the Kyoto group [15], who considered a memory with a single loading and a single erasing channel. However, such memory is not fully reliable, and in exceptional cases, the erasing pulse does not enter the ring, so the memory state does not change. The construction and experimental verification of a more reliable memory ring with two erasing channels was reported in [16,17,18]. Still, the geometry of ring and channels require a high precision to make the memory operational. The idea of memory based on a ring in excitable continuous medium can be adapted to discrete interacting oscillators arranged in a circular geometry [19]. In the simplest case, memory is assembled using oscillators that can be individually inhibited. Experiments with three droplets containing reagents of a photosensitive Belousov–Zhabotinsky (BZ) reaction [20,21,22] demonstrated that in such a system, pulses propagating in clockwise and counterclockwise direction code stable memory states [23]. Moreover, the state of memory can be changed by droplet-specific illumination.

In the reports listed above, the memory states were coded in dynamically changing concentration profiles. Here we discuss another approach to chemical memory based on discrete Turing patterns. Our approach follows the idea of information processing with networks of interacting chemical reactors introduced in [24,25,26]. In these papers, the authors focused on nodes that show excitable or bistable behavior; thus, the concentration of reagents in a single node can evolve toward one of two values that can be interpreted as corresponding to binary logical values [27]. It has been demonstrated that reaction networks can perform logic gate operations or act as binary memory. The idea of computing with oscillator networks proposed in [28] generalizes the above mentioned approach. It has been shown that oscillator networks can perform highly accurate non-trivial classification tasks, including those medically oriented [29,30]. It was found that for specific reactions, the networks can have stable discrete Turing patterns; thus, they can function as a chemical memory [31,32,33]. Reaction–diffusion patterns in a ring of twenty coupled cells representing a continuous tissue were defined by Alan Turing as a key factor of morphogenesis [34]. The driving force for the spontaneous occurrence of Turing patterns is the difference in diffusion coefficients between activator and inhibitor; specifically, the inhibitor transport rate has to be faster than the activator transport rate. This condition is called long-range inhibition short-range activation [35,36]. The requirement for Turing instability is one of the critical elements of yeast budding [37,38]. It seems to be a key factor for cerebral cortex development during fetal state [39], and it might be behind brain abnormalities [40]. It was later shown by Wolpert et al. [41] that the spatiotemporal patterns occurring during morphogenesis do not always obey Turing’s conditions for instability, involving the transport rates of activating and inhibiting molecules.

The discrete Turing patterns and oscillations can be observed in the same system as shown by Bar-Eli [42]. The system of coupled cells spontaneously oscillates even when special conditions for Turing instability under an equal transport coefficients are met. One can toggle in between Turing patterns and oscillations using carefully targeted perturbations [43]. Early experiments with BZ reaction systems showed oscillation death regimes (discrete Turing patterns) in reactors coupled by peristaltic pumps [44,45,46] or different types of valves [47] between adjacent reactors instead of cellular membranes. The experimental findings in terms of transitions between oscillations and discrete non-uniform patterns were supported by models of two, four, and twenty coupled cells [31,32,48]. These transitions represented the basics of our chemical computing techniques and were therefore used also for three and four-coupled cells with various layouts to work as multiargument logic operations [33,48]. Here we are concerned with testing these patterns as potential memory states.

The paper is organized as follows. In Section 2, we provide details of the glycolytic model and formulate equations describing arrays of three coupled cells oriented in the triangular geometry that is supposed to work as a chemical memory. Moreover, we discuss the stability and bifurcations of the stationary states and identify parameter values for which the system has a number of stable discrete Turing patterns that are used to code symbols. In Section 3, we discuss how to write a specific symbol for the memory and how the memory can be erased. Our conclusions and plans for future work are formulated in the Discussion.

## 2. Methods

### The Model for Glycolysis

The glycolytic oscillatory reaction is one of the most important biochemical processes. In mammalian cells, it can be divided into aerobic and anaerobic glycolysis. While aerobic glycolysis ends with pyruvate and the rest is conducted in mitochondria, anaerobic glycolysis continues to work with pyruvate until it forms lactate [49] or even ethanol [50]. The special form of aerobic glycolysis with high affinity to lactate is called the Warburg effect [51,52]. The glycolysis itself has morphogenetic effects [53], and its high glucose uptake can also be used to simulate cerebral cortex pattern formation [40,54,55]. Glycolysis in cells of Saccharomices cerevisiae can show oscillations [56,57], bistability [58,59], excitation and birhytmicity [60]. Furthermore, neuron cells and yeast cells share similarities, which can be used to study neurological diseases via neural yeast models [61]. These similarities also implicate the use of yeast or yeast extract for chemical computing as logic gates [62].

Here we consider applications of coupled reactors in which glycolytic oscillatory reaction proceeds as a chemically computing medium. We investigate the system of three coupled reactors illustrated in Figure 1 as a candidate for chemical memory. The coupling between nodes is provided by the exchange of reagents. The model used in our simulations is based on the core glycolytic kinetics proposed by Moran and Goldbeter [60]. It describes the time evolution of the network by time-dependent concentrations of ATP and ADP at all nodes. In order to minimize the numerical complexity of the problem and to neglect reaction steps not responsible for dynamic behavior, we selected the model with concentrations of just two reactants, namely ATP (the variable x(t) denotes its time-dependent concentration) and ADP (described by y(t)). The same simple model was used in a few papers on binary logic gates operating on the information coded in discrete Turing patterns [32,33,48]. The progress of glycolytic reaction proceeding in a continuously stirred reactor (CSTR) is described with the following equations: (1)$$\begin{array}{c} \frac{dx}{dt}=F_{x}\left(x\left(t\right),y\left(t\right),\nu \left(t\right)\right)=\nu \left(t\right)+\sigma_{inh}\cdot \frac{y{\left(t\right)}^{n}}{M^{n}+y{\left(t\right)}^{n}}-\sigma_{M}\cdot \frac{x\left(t\right)*\left(1+x\left(t\right)\right)*{\left(1+y\left(t\right)\right)}^{2}}{L+{\left(1+x\left(t\right)\right)}^{2}*{\left(1+y\left(t\right)\right)}^{2}}, \end{array}$$(2)$$\begin{array}{c} \frac{dy}{dt}=F_{y}\left(x\left(t\right),y\left(t\right)\right)=\phi \sigma_{M}\cdot \frac{x\left(t\right)*\left(1+x\left(t\right)\right)*{\left(1+y\left(t\right)\right)}^{2}}{L+{\left(1+x\left(t\right)\right)}^{2}*{\left(1+y\left(t\right)\right)}^{2}}-\phi \sigma_{inh}\cdot \frac{y{\left(t\right)}^{n}}{M^{n}+y{\left(t\right)}^{n}}-k_{s}*y\left(t\right). \end{array}$$Here ν(t) describes the ATP inflow rate, $\sigma_{inh}$ is the inhibition rate coefficient, *M* is the Michaelis constant, *n* gives the Hill coefficient, *L* is the allosteric constant representing affinity of the PFK to its reactive conformation R rather than to its non-reactive conformation L [63], $\sigma_{M}$ denotes the autocatalysis rate coefficient, ϕ is the ratio between dissociation constants of ATP to ADP, and, finally, $k_{s}$ represents the rate coefficient of ADP degradation. In simulations, we used the same values of model parameters as reported in other papers that considered the same model [33,60]: n=4, M=10, $L=5.0\times 10^{6}$, ϕ=1 and $k_{s}=0.06\;s^{-1}$. Specifically for discrete Turing patterns, ν has to be in one of two parameter regions: either $\nu \;\approx \;0.22\pm 0.01\;s^{-1}$ or $\nu \in (1.02\;s^{-1},1.92\;s^{-1})$. The selection of $\sigma_{inh}$ and $\sigma_{M}$ values is discussed at the end of the section.

As a potential chemical memory device, we investigate a small network of three nodes that represent continuously stirred reactors with the time evolution described by the above equations. The network geometry is illustrated in Figure 1. Big circles represent network nodes. The node color is used to plot time-dependent concentrations of reagents in the following figures. The colored arrows directed toward the nodes mark time-dependent inflows of ATP (the first term on the right side of Equation (Equation 1)). Following the previous studies on interconnected glycolytic reactors [32,33,48], we assume that the nodes interact due to flows of reagents between them. We consider a fully symmetric network with identical flows between nodes with the rate $k_{D}$. The equations describing network evolution have the form: (3)$$\begin{array}{c} \frac{dx_{j}}{dt}=F_{x}\left(x_{j}\left(t\right),y_{j}\left(t\right),\nu_{j}\left(t\right)\right)+k_{D}\cdot \sum_{i=1,i\neq j}^{i=3}\left(x_{i}\left(t\right)-x_{j}\left(t\right)\right), \end{array}$$(4)$$\begin{array}{c} \frac{dy_{j}}{dt}=F_{y}\left(x_{j}\left(t\right),y_{j}\left(t\right)\right)+k_{D}\cdot \sum_{i=1,i\neq j}^{i=3}\left(y_{i}\left(t\right)-y_{j}\left(t\right)\right). \end{array}$$where symbols *i* and *j* index the nodes. In our simulations, we followed previous studies on discrete Turing patterns in systems with glycolytic reactions and used $k_{D}=0.1\;s^{-1}$ [32,33,48].

Figure 2 illustrates the bifurcation diagram for the considered network in the phase space of parameters $\sigma_{inh}\in \left[40\;s^{-1},100\;s^{-1}\right]$ and $\sigma_{M}\in \left[100\;s^{-1},250\;s^{-1}\right]$. The lines separate the phase space into regions with different numbers of stationary states. The symbols N−M indicate that there are *N* stationary states in the given region of $\left(\sigma_{inh},\sigma_{M}\right)$ of which *M* are stable. We used the following line coding: the red curves are the Hopf bifurcation lines, blue curves mark symmetry-breaking bifurcation lines, and green curves are the limit point lines. The solid and dashed lines are used to indicate a potential change of stability at the line crossing. There is no change of stability between regions separated by a dashed line, whereas if regions are separated by a solid line, such change occurs. To obtain the bifurcation diagram, we set $\nu_{i}\left(t\right)\equiv 1.84$ for i=1,3 because such values of ATP inflow rates were frequently used in the previous reports.

For the discussion on the potential usefulness of discrete Turing pattern as a chemical memory, we selected $\sigma_{inh}=80\;s^{-1}$ and $\sigma_{M}=200\;s^{-1}$ because, for this pair of rates, the system shows six stable discrete patterns. We believe such rates apply to a system with hyperthermophilic bacterium Thermotoga maritima, which was cloned and functionally expressed in Escherichia coli and operates at temperatures reaching up to 100 °C [64].

## 3. Results

Let us discuss the application of the network as a chemical memory. For the selected set of parameters, the homogeneous stationary state, characterized by concentrations $x_{1}=x_{2}=x_{3}\;\approx \;58.07$ and $y_{1}=y_{2}=y_{3}\;\approx \;30.67$, is unstable. After its perturbation, the system usually approaches the stable limit cycle characterized by the period ∼48.2 s. The amplitudes of x− and y− oscillations are ∼87.2 and ∼89.8, respectively. We assume that the limit cycle corresponds to the empty memory in which no information is coded. It is like a clear page on which information can be written in. When we allow for the symmetry breaking then, the system shows two classes of stable discrete Turing patterns. One of them is formed by three patterns: 


#1 characterized by $x_{1}\;\approx \;30.93$, $x_{2}=x_{3}\;\approx \;97.82$ and $y_{1}\;\approx \;67.83$, $y_{2}=y_{3}\;\approx \;12.08$,#2 characterized by $x_{1}=x_{3}\;\approx \;97.82$, $x_{2}\;\approx \;30.93$ and $y_{1}=y_{3}\;\approx \;12.08$, $y_{2}\;\approx \;67.83$,#3 characterized by $x_{1}=x_{2}\;\approx \;97.82$, $x_{3}\;\approx \;30.93$ and $y_{1}=y_{2}\;\approx \;12.08$, $y_{3}\;\approx \;67.83$, In the following, we assume these patterns code symbols A, B, and C, respectively. The discrete Turing pattern representing the A symbol is shown in Figure 3a. The solid and dashed lines illustrate stationary concentrations of ATP and ADP.


Another class of stable stationary states includes the patterns:


#4 characterized by $x_{1}\;\approx \;84.49$, $x_{2}=x_{3}\;\approx \;45.28$ and $y_{1}\;\approx \;8.88$, $y_{2}=y_{3}\;\approx \;41.56$,#5 characterized by $x_{1}=x_{3}\;\approx \;45.28$, $x_{2}\;\approx \;84.49$ and $y_{1}=y_{3}\;\approx \;41.56$, $y_{2}\;\approx \;8.88$,#6 characterized by $x_{1}=x_{2}\;\approx \;45.28$, $x_{3}\;\approx \;84.49$ and $y_{1}=y_{2}\;\approx \;41.56$, $y_{3}\;\approx \;8.88$, We assume these patterns code symbols X, Y, and Z, respectively. The discrete Turing pattern corresponding to symbol Z is illustrated in Figure 4a. The solid and dashed lines mark stationary concentrations of ATP and ADP.


The system also has a class of six unstable stationary patterns that can be generated by permutations of the following concentrations: $x_{1}\;\approx \;87.618,\;x_{2}\;\approx \;95.706,\;x_{3}\;\approx \;32.449,$
$\;y_{1}\;\approx \;17.588,\;y_{2}\;\approx \;10.849$, and $y_{3}\;\approx \;63.563$. All stationary states listed above make 13-6 states that should be seen in the region where the asterisk is located (cf. Figure 2).

Now let us illustrate the usefulness of the network of CSTRs as a chemical memory by demonstrating that it can be easily written and erased. As the first step, we show that the memory loaded with any symbol listed above can be easily switched to the oscillating mode by changing the ATP inflow. To do this, we consider time-dependent inflows in the form:(5)$$\nu_{i}\left(t\right)=\nu_{0}+\frac{\alpha }{\left(1+\exp\left(-\beta \left(t-t_{0}\right)\right)\right)((1+\exp\left(\beta \left(t-\left(t_{0}+dt\right)\right)\right)}.$$Here $\nu_{0}$ is the constant component of the inflow (in our simulations $\nu_{0}=1.84\;s^{-1}$), and the second term represents its step-like perturbation characterized by the amplitude α, the initiation time $t_{0}$, width dt, and the steepness β. Of course, if the perturbation is applied to a stable state, the value of $t_{0}$ does not change the outcome but just indicates when the action happens.

The rectangular form of state-changing perturbation has been suggested by the results presented in [48]. Here we considered only perturbations that increased the ATP inflows if compared to the conditions at which six discrete Turing patterns were observed. We did not develop a systematic algorithm for scanning the space of perturbations to find α and dt characterizing inflows at individual nodes. We performed a random parameter search for perturbations that produce the required transformation of symbols. If such perturbation was identified, we manually checked if its amplitude and width could be reduced without losing its function. The results obtained using such a procedure are given below.

The process of erasing memory loaded with the *A* symbol is illustrated in Figure 3. The inflow perturbation described by the second term of Equation (Equation 5) is characterized by $\alpha =0.1\;s^{-1}$, $t_{0}=500$, dt=200, and β=0.2 and is plotted with the black line Figure 3b. It is applied to all nodes of the network. The sub-figures (c,d), (e,f), and (g,h) of Figure 3 show the time evolution x(t) and y(t) in the nodes 1, 2, and 3, respectively before, during, and after the perturbation was applied. After the perturbation, the network reaches the limit cycle. Having in mind the network symmetry and the symmetry of applied perturbation, the inflow perturbation shown in Figure 3b transforms both the memory states *B* and *C* into the blank memory.

As seen, the inflow perturbation is small, and its amplitude is around 5% of the ATP stationary inflow that stabilizes the nonlinear behavior of glycolytic reaction. Moreover, the perturbation acts for a short time; that is, as long as a 5 oscillation period. It is worth mentioning that the illustrated perturbation includes some optimization because if the value of α is reduced to $0.05\;s^{-1}$ or if dt is reduced to 100, then the memory in states *A*, *B*, or *C* does not change.

The process of erasing memory loaded with the *Z* symbol is illustrated in Figure 4. The perturbation described by the second term of Equation (Equation 5) is characterized by $\alpha =1.0\;s^{-1}$, $t_{0}=500$, dt=100, and β=0.2 and is plotted with the black line Figure 4b. It is applied to all nodes of the network. The sub-figures (c,d), (e,f), and (g,h) of Figure 4 show the time evolution x(t) and y(t) before, during, and after the perturbation in the nodes 1, 2 and 3, respectively. As in the erasing of *A* symbol, after the perturbation is applied, the network reaches the limit cycle. Due to problem symmetry, the perturbation shown in Figure 4b transforms both the memory states *X* and *Y* into the blank memory. The parameters of presented perturbation are optimized because if the value of α is reduced to $0.8\;s^{-1}$ or if dt is reduced to 50 then the memory states *X*, *Y*, and *Z* remain in these states after the perturbation ends.

The results of Figure 3 and Figure 4 induce the question if we can have a perturbation erasing all six memory states. The positive answer is straightforward: the perturbation with maximum values parameters that characterize both perturbations illustrated above (i.e., $\alpha =1.0\;s^{-1}$, $t_{0}=500$, dt=200, and β=0.2) transforms any discrete Turing pattern into the limit circle. It is also worth mentioning that these perturbations do not write any symbol if applied to the blank memory.

Let us consider the problem of memory writing. Now perturbation should act differently on different nodes. Writing one of the symbols *A*, *B*, or *C* is easy. The evolution of concentrations when writing the *B* symbol into the blank memory is illustrated in Figure 5. The pattern corresponding to this state is characterized by a low *y* value (∼12.08) in nodes #1 and #3 and a high *y* value in node #2 (∼67.83). We can obtain such a state (cf. Figure 5a) if the inflows for nodes #1 and #3 are unperturbed and inflow perturbation defined by Equation 5 with parameters ($\alpha =2.8\;s^{-1}$, $t_{0}=500$, dt=100 and β=0.2.) is applied to the node #2. In Figure 5, the pairs of sub-figures (b,c), (d,e), and (f,g) show the time evolution of x(t) and y(t) in all nodes before, during, and after the perturbation. We confirm that the perturbation presented in Figure 5a transforms the limit cycle to the memory state *B* irrespectively on the limit cycle phase when it is applied. Our argument is based on simulations in which the perturbation was applied to 50 equally time-distributed points on the cycle, and in every case, the memory evolved to the state *B*. It can be verified that perturbations characterized by the same parameters applied to the nodes #1 and #3 respectively write the symbols *A* and *C* into the blank memory.

Writing symbols *X*, *Y*, and *Z* into the blank memory requires more complex perturbation involving two nodes. An example illustrating the process of writing *Z* symbol into the blank memory is shown in Figure 6. Here the inflow perturbations are applied to the node #1 (the blue line presents $\nu_{1}\left(t\right)$ characterized by $\alpha =0.5\;s^{-1}$, $t_{0}=500$, dt=200, and β=0.2) and to the node #2 ($\nu_{2}\left(t\right)$ defined by $\alpha =1.9\;s^{-1}$, $t_{0}=500$, dt=700, and β=0.2 is plotted using the green line). The inflow at the node #3 is unperturbed ($\nu_{3}\left(t\right)\equiv 1.84\;s^{-1}$). The pairs of sub-figures (b,c), (d,e), and (f,g) show the time evolution of x(t) and y(t) in all nodes before, during, and after the perturbation. After the applied perturbations, the node for which the ATP inflow remained unperturbed is characterized by the low value of stationary ATP concentration. We confirmed that the symbol *Z* is written to the memory irrespective of the phase of the limit cycle at the moment when the considered perturbation is applied. The same strategy can be used to write the *X* and *Y* symbols into the blank memory.

The presented results show that one can easily change the state of loaded memory from one symbol to another. This can be achieved by erasing the old symbol and writing the new one into the blank memory.

## 4. Discussion

The presented simulation results confirm the usefulness of discrete Turing patterns as a chemical memory. We considered a small system composed of three nonlinear nodes and selected the adjustable parameters of the glycolytic reaction model such that the system shows stable homogeneous oscillations as well as six different stable non-homogeneous concentration patterns. The system oscillations are interpreted as blank memory, and each pattern represents another symbol written into it. We found that the memory can be erased by a short perturbation of the ATP inflow and identified the inflow perturbations that transform oscillations into stable patterns. Two glycolytic nodes were needed to store 1 bit of information (two symbols) in previously discussed realizations of a chemical memory [32,33]. Therefore, coding $2^{R}$ symbols (*R* bit string) requires *R* binary networks. The three-node network discussed in this paper allows for a higher density of data storage. We can code $6^{R}=2^{\log_{2}\left(6\right)R}$ symbols that contain $\log_{2}\left(6\right)R≅2.58R$ bits of information if we consider *R* three-node networks. Thus, the memory based on three-node networks requires $3\log_{6}\left(2\right)≅1.16$ nodes for coding a single bit.

It may be expected that systems with a larger number of nodes can produce a yet higher density of memory states. However, the published bifurcation diagrams for networks characterized by four nodes and by different geometries of interactions [48] do not show sets of parameters with more than six stable Turing patterns. Therefore, further investigation on the number of stable patterns for networks with a large number of nodes is necessary.

We focused our attention on symbol coding with stable Turing patterns because reading out such information seems simpler than extracting information from a stable dynamic pattern. It can be anticipated that the number of dynamic patterns can be higher and allow for denser symbol coding. For example, eight stable oscillation modes were observed in experiments with three interacting droplets containing a solution of BZ reaction reagents [23]. It can be expected that in networks composed of a large number of nodes, both static and dynamic patterns can be used for symbol coding. If so, the time needed to extract information from a specific state can play an important role.

In the common meaning, the memory contains information on our previous experiences. In the considered system, the “experience” is hidden in the function ν(t). It would be interesting to identify the variables that parameterize this function and relate them to the memory state of the system after a certain type of “life” perturbation is applied.

Studies on chemical memory are important because they provide insights into the fundamental principles that govern the behavior of complex chemical systems. They can have potential applications in fields such as materials science or drug delivery, where they could be used to create smart materials or devices that can sense and specifically respond to changes in their environment according to previously learned strategies.

## Figures and Tables

**Figure 1 biomimetics-08-00154-f001:**
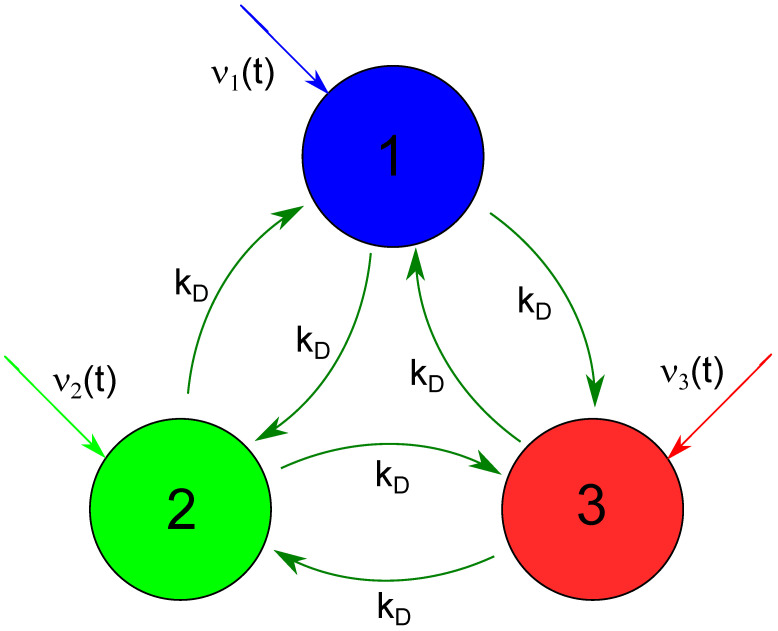
The geometry of a network considered as a chemical memory. Big circles represent network nodes (CSTRs) where the glycolytic reaction proceeds. The following figures use the same node color to plot time-dependent concentrations of reagents. Dark green arrows interlinking nodes illustrate reagent flows represented by the last two terms on Equations (3) and (4). Colored arrows directed toward the nodes mark time-dependent inflows of ATP (the first term on the right side of Equation (Equation 1)).

**Figure 2 biomimetics-08-00154-f002:**
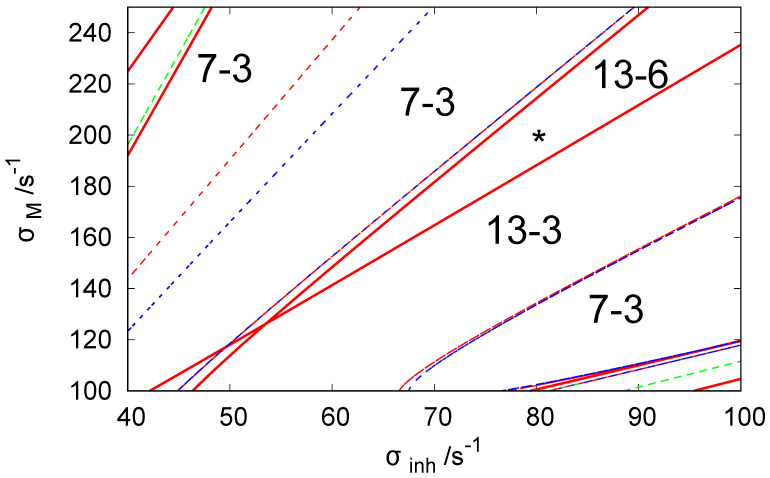
The bifurcation diagram for three coupled CSTRs with a cyclic geometry of connections illustrated in Figure 1 in the phase space of parameters $\sigma_{inh}$ and $\sigma_{M}$. The system of Equations (3) and (4) is solved with $k_{D}=0.1\;s^{-1}$. The line coding: Red curve—the Hopf bifurcation line, blue curve— symmetry breaking bifurcation line, green curve—limit point curve/line. The solid line indicates the change of stability at the line crossing. The is no change of stability between regions separated by a dashed line. The notation N−M indicates that there are *N* stationary states in the region of the phase space and *M* of these states are stable. The asterisk symbol marks the location of parameter values used in our simulations of chemical memory ($\sigma_{inh}=80\;s^{-1}$ and $\sigma_{M}=200\;s^{-1}$).

**Figure 3 biomimetics-08-00154-f003:**
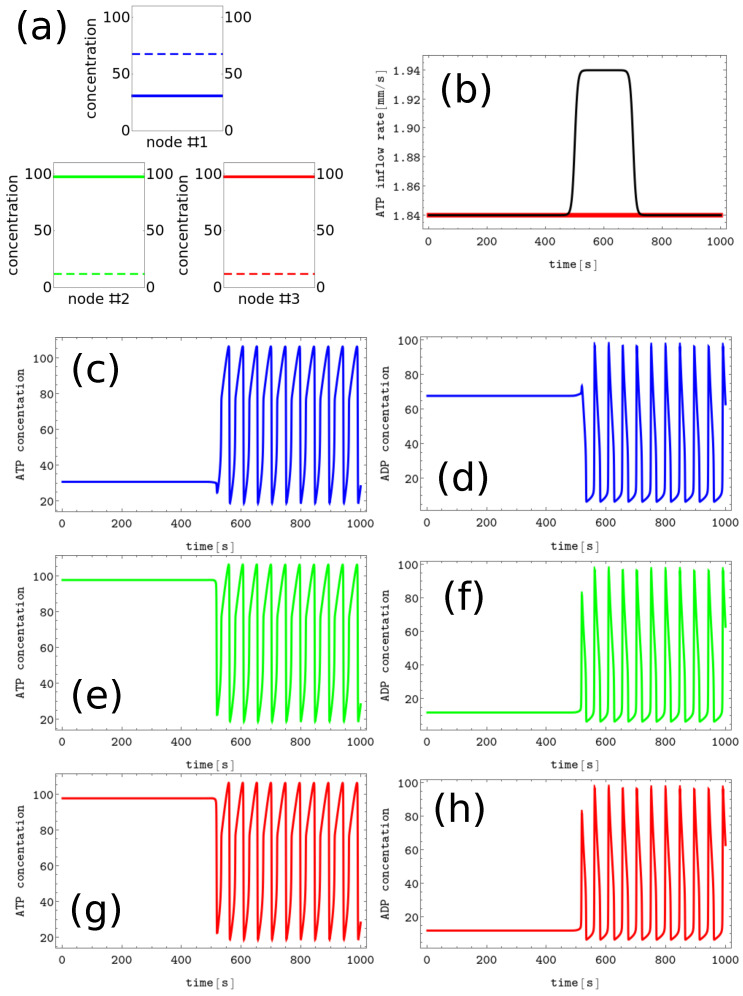
The process of erasing memory loaded with the *A* symbol. (**a**) The structure of concentration values in the discrete Turing pattern corresponding to symbol A. The solid and dashed lines represent concentrations of ATP and ADP. (**b**) The inflow combined with perturbation applied to all nodes $\nu_{i}\left(t\right),i=1,\;3$ (the black curve, $\alpha =0.1\;s^{-1}$, $t_{0}=500$, dt=200 and β=0.2) compared with the inflow characterizing unperturbed memory (the red curve $\nu_{0}\equiv 1.84\;s^{-1}$). The pairs of sub-figures (**c**–**h**) show the time evolution of x(t) and y(t) in all nodes before, during, and after the perturbation. The color coding is the same as in Figure 1.

**Figure 4 biomimetics-08-00154-f004:**
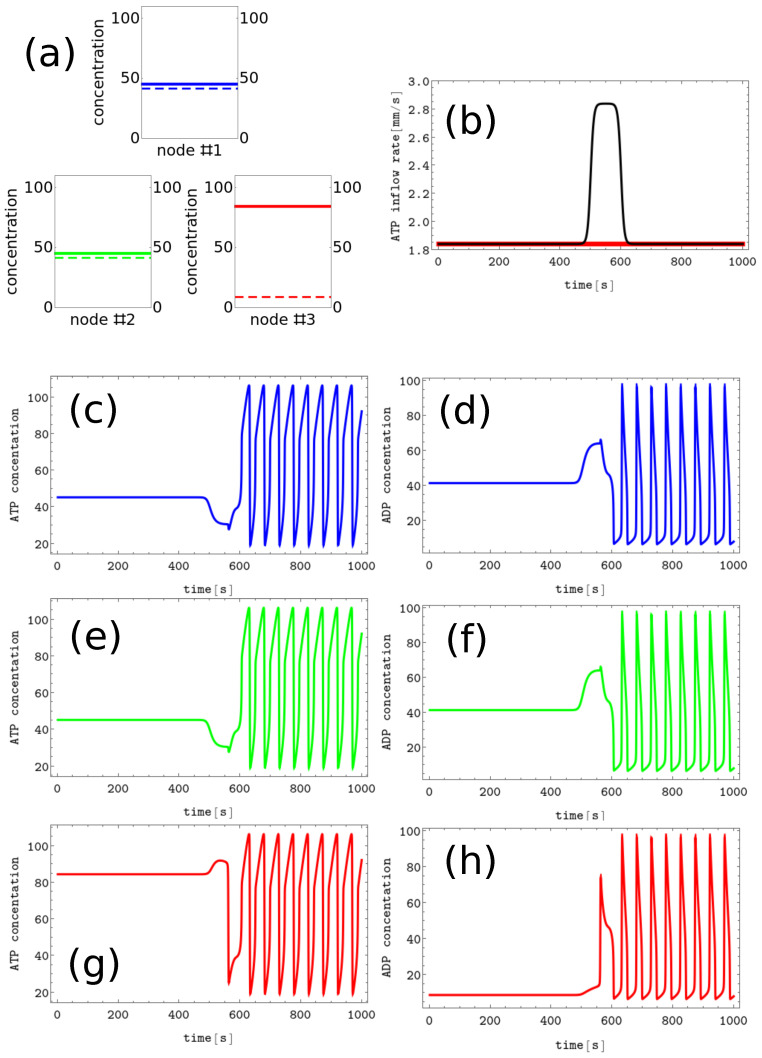
The process of erasing memory loaded with the *Z* symbol. (**a**) The structure of concentration values in the discrete Turing pattern corresponding to symbol Z. The solid and dashed lines represent concentrations of ATP and ADP. (**b**) The inflow combined with perturbation applied to all nodes $\nu_{i}\left(t\right),i=1,\;3$ (the black curve, $\alpha =1.0\;s^{-1}$, $t_{0}=500$, dt=100, and β=0.2) compared with the inflow characterizing unperturbed memory (the red curve $\nu_{0}\equiv 1.84\;s^{-1}$). The pairs of sub-figures (**c**–**h**) show the time evolution of x(t) and y(t) in all nodes before, during, and after the perturbation. The color coding is the same as in Figure 1.

**Figure 5 biomimetics-08-00154-f005:**
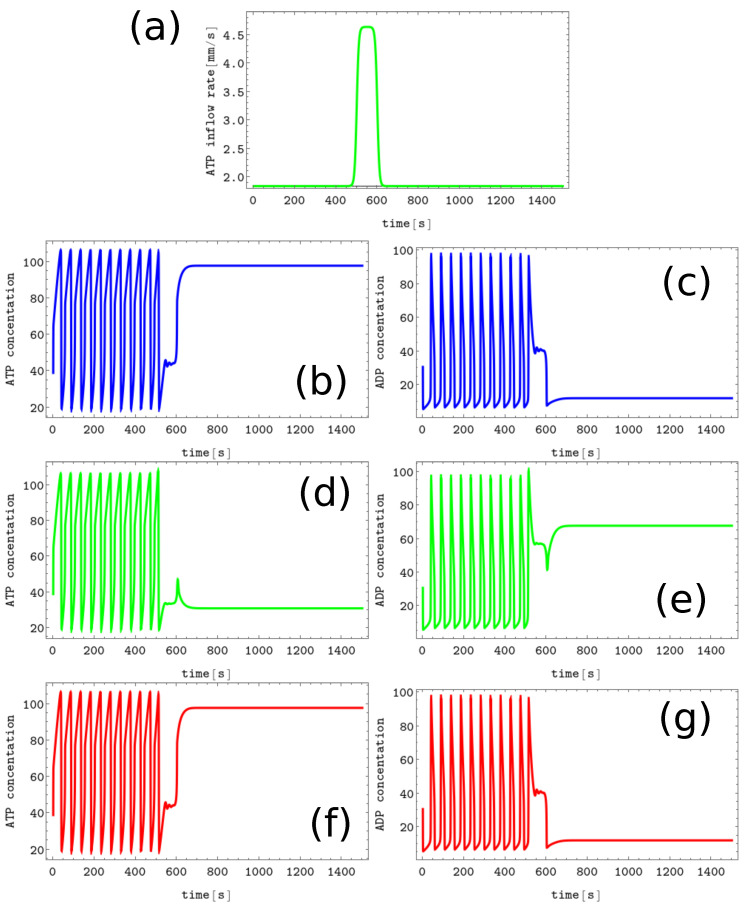
The process of writing the *B* symbol into the blank memory. (**a**) The black line shows unperturbed inflow at nodes #1 and #3 ($\nu_{1}\left(t\right)=\nu_{3}\left(t\right)\equiv 1.84\;s^{-1}$), the green curve illustrates $\nu_{2}\left(t\right)$ ($\alpha =2.8\;s^{-1}$, $t_{0}=500$, dt=100, and β=0.2). The pairs of sub-figures (**b**–**g**) show the time evolution of x(t) and y(t) in all nodes before, during, and after the perturbation. The color coding is the same as in Figure 1.

**Figure 6 biomimetics-08-00154-f006:**
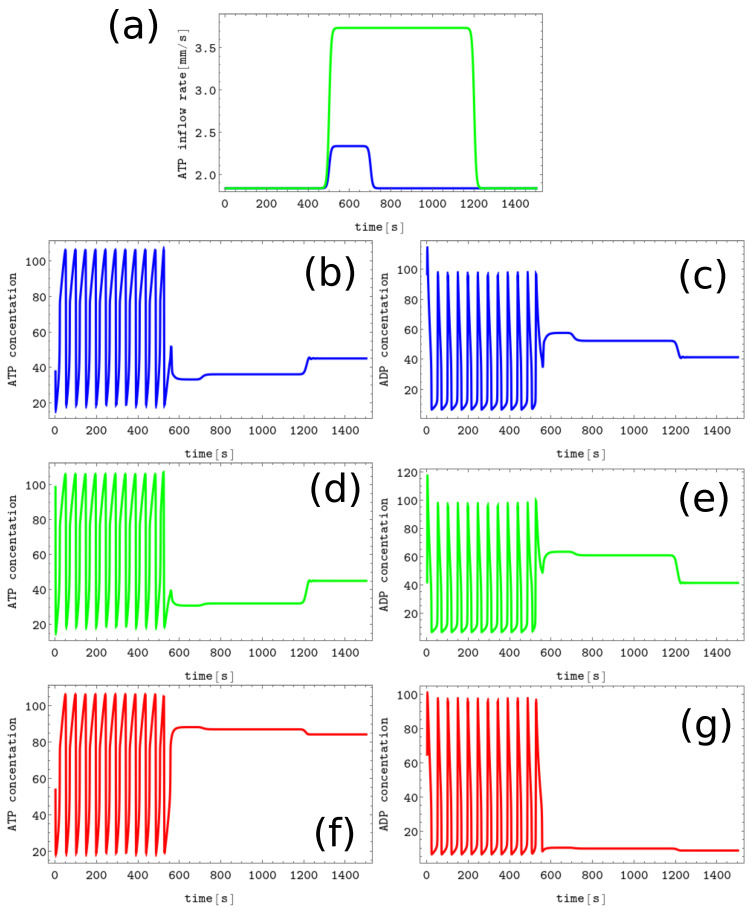
The process of writing the *Z* symbol into the blank memory. (**a**) The blue line shows $\nu_{1}\left(t\right)$ ($\alpha =0.5\;s^{-1}$, $t_{0}=500$, dt=200, and β=0.2), the green line presents $\nu_{2}\left(t\right)$ ($\alpha =1.9\;s^{-1}$, $t_{0}=500$, dt=700 and β=0.2), the inflow at the node #3 is unperturbed ($\nu_{3}\left(t\right)\equiv 1.84\;s^{-1}$). The pairs of sub-figures (**b**–**g**) show the time evolution of x(t) and y(t) in all nodes before, during, and after the perturbation. The color coding is the same as in Figure 1.

## Data Availability

Raw data and computer programs that generated presented results are available in the public repository at https://doi.org/10.18150/EWF30H (accessed on 1 March 2023) or from the corresponding author after contact.

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
