# Peer review of "Chemical Memory with Discrete Turing Patterns Appearing in the Glycolytic Reaction"

_biomimetics, 2023, doi:10.3390/biomimetics8020154_

Round 1

Reviewer 1 Report

The manuscript by Gorecki and Muzika addresses an important field of unconventional computing - memory formation in dynamic chemical systems.

All the numerical simulations are well described, results are clearly presented and conclusions are sound.

There is only one minor point that may need more detailed discussion: Memory writting and erasure reqiures well-defined inflow of ATP, described by Eq. 5. For some cases the parameters for Eq. 5 are given, however it seems from the manuscript that there is no well-defined protocol of determination of parameters for each inflow "pulse". A more detailed discussion of the desing of writting/erasing inflow pulses will greatly increase the overall merit of the paper.

Therefore I recommend the manuscript for publication upon minor revision.

Author Response

We are grateful for the comment. The information on how the parameters of the state-changing perturbation were found has been included in a new paragraph below Eq.5. We wrote:

The rectangular form of state-changing perturbation has been suggested by the results presented in \cite{Muzika 2020}. Here we considered only perturbations that increased the ATP inflows if compared to the conditions at which 6 discrete Turing patterns were observed. We did not develop a systematic algorithm for scanning the space of perturbations to find $\alpha$ and $dt$ characterizing inflows at individual nodes. We performed a random parameter search for perturbations that produce the required transformation of symbols. If such perturbation was identified, we checked manually if its amplitude and width could be reduced without losing its function. The results obtained using such a procedure are given below.

Reviewer 2 Report

The authors show in their manuscript that 3 coupled CSTRs with glycolysis can be used as memory units. A 2-variable model with ATP and ADP developed earlier is utilized to model the reaction. The parameters are chosen in such a way that Turing patterns can appear. The bifurcation diagram illustrates that at the chosen parameters 13 stationary states exist from which six are stables. The memory erasure is established as oscillatory patterns while the writing in the blank memory appears as stable stationary concentrations (oscillations disappear).

The modelling is appropriately described, the figures are nice.

Turing patterns are spatial structures therefore I miss that at least one figure should show the spatial discrete Turing structure. Always the time evolution of the concentrations of the two species in the  3 CSTRs are plotted in different graphs. The authors should explain in one figure what is meant by the discrete Turing patterns in their manuscript and the rest of the figures can be left as submitted by the authors.

Typos left in the manuscript:

pg.2 line 88: con toggle -> can toggle

pg. 2 line 91: in between adjacent reactors -> between ...

Based on above I recommend the manuscript for publication after minor revision.

Author Response

We are grateful for the comments.

The typos have been corrected, thanks for pointing them!

What concerns the discrete Turing patterns we have illustrated the patterns corresponding to symbols A and Z in newly added Figs. 3a and 4a. The description of these sub-figures is in the captions and in the text after the patterns are defined. The added text is marked red.